# Frequency of Invasive Fungal Disease in Adults: Experience of a Specialized Laboratory in Medellín, Colombia (2009–2015)

**DOI:** 10.3390/jof6010039

**Published:** 2020-03-20

**Authors:** Yorlady Valencia, Diego H. Cáceres, Catalina de Bedout, Luz E. Cano, Ángela Restrepo

**Affiliations:** 1Medical and Experimental Mycology Unit, Corporación para Investigaciones Biológicas (CIB), Medellín 050036, Colombia; yorly927@hotmail.com (Y.V.); catybedout@gmail.com (C.d.B.); lcano@cib.org.co (L.E.C.); angelares@une.net.co (Á.R.); 2Mycotic Diseases Branch, Centers for Disease Control and Prevention (CDC), Atlanta, GA 30333, USA; 3Center of Expertise in Mycology Radboudumc/CWZ, 6525GA Nijmegen, The Netherlands; 4School of Microbiology, University of Antioquia, Medellín 050036, Colombia

**Keywords:** mycoses, laboratory methods, histoplasmosis, aspergillosis, cryptococcosis, pneumocystosis, paracoccidioidomycosis, invasive candidiasis

## Abstract

Invasive fungal diseases (IFD) contribute significantly to worldwide morbidity and mortality, but their frequency is not well-described in some countries. The present work describes the frequency of IFD in a specialized laboratory in Colombia. A retrospective, descriptive study was implemented between March 2009 and December 2015. Results: 13,071 patients with clinical suspicion of IFD were referred during the study period, from which 33,516 biological samples were processed and analyzed using 14 laboratory methods. Diagnosis was confirmed in 1425 patients (11%), distributed according to the mycoses of interest analyzed here: histoplasmosis in 641/11,756 patients (6%), aspergillosis in 331/10,985 patients (3%), cryptococcosis in 239/8172 patients (3%), pneumocystosis in 111/1651 patients (7%), paracoccidioidomycosis in 60/10,178 patients (0.6%), and invasive candidiasis in 48/7525 patients (0.6%). From the first year of the study period to the last year, there was a 53% increase in the number of cases of IFD diagnosed. Our laboratory experienced a high frequency of IFD diagnosis, possibly attributable to the availability of a greater range of diagnostic tools. Frequency of IFD in this study was atypical compared with other studies, probably as a result of the single laboratory-site analysis. This demonstrates that implementing educational strategies helps to create a high index of clinical suspicion, while the availability and utilization of appropriate diagnostic assays assure greater reliability in identification of these cases.

## 1. Introduction

Invasive fungal diseases (IFD) are an important cause of worldwide morbidity and mortality. Therefore, their prevalence is subject to study by GAFFI (Global Action Fund for Fungal Infections). To date, GAFFI strategy has been implemented in 55 countries, but data from Colombia is limited [1]. Incidence of IFD has increased significantly in the last three decades due to an increase in the population at risk (i.e., patients who are transplant recipients of hematopoietic progenitors or solid organs, users of immunosuppressive treatments, and those with advanced HIV disease and other autoimmune diseases). These individuals have a greater susceptibility to opportunistic fungal infections caused by various fungal genera (*Candida, Aspergillus, Histoplasma, Cryptococcus, Talaromyces, Emergomyces, Pneumocystis, Mucorales, Fusarium, Scedosporium*, etc.), many of which are more virulent and frequently associated with antifungal resistance [2,3,4,5].

Endemic mycoses caused by fungi of the genera *Histoplasma, Paracoccidioides, Coccidioides,* and *Blastomyces* may be related to changes in the habits of the population, such as occupational activities (e.g., agriculture, livestock), leisure, tourism and migration into the microorganism’s possible natural habitat. However due to the prolonged latency of the infection, these diseases have been reported in regions where they are considered non-endemic [3]. Globally, fungal infections are not considered a priority public health problem and are non-notifiable, which contributes to the underreporting of these diseases. There is also low clinical suspicion among physicians, who infrequently include fungal infections in their differential diagnoses. Additionally, some of these diseases are lacking of standardized and validated diagnostic methods, and laboratories with the capacity to diagnose IFD are limited [1].

Diagnosis of IFD is done using many different laboratory methods. For most of these diseases, the gold standard for diagnosis is based on conventional laboratory assays using culture or histopathology (including special stains). Yet these assays present several limitations, exposing the need for high-level laboratory infrastructure for cultures handling (biosecurity level 3) and highly trained laboratory staff. These assays also presented variable analytical performance, and long turn-around time for laboratory tests results [6]. Other alternatives for IFD include assays for the detection of specific antibodies, antigens, and fungal DNA. These alternative assays present higher analytical performance and lower turn-around time for results, compared to conventional laboratory assays [6].

Some studies have described the epidemiology of IFD in Latin America. In 2016, Giacomazzi et al. [7] based on Brazilian government data for 2011, and data extrapolated from the literature estimated the approximate incidence of severe fungal infections per 1000 hospital admissions. Authors reported values of 7.9; 7.1; and 2.0 cases/1000 admissions for paracoccidioidomycosis (PCM), coccidioidomycosis, and histoplasmosis, respectively. In another study published in Guatemala in 2017, the authors reported the following data for the 2013–2015 period, 16,695 cases of oral candidiasis, bronchopulmonary aspergillosis in 5568 asthmatic adults, 4505 with esophageal candidiasis, 816 cases of *Pneumocystis* pneumonia, 705 cases of disseminated histoplasmosis, 495 individuals with chronic pulmonary aspergillosis, and 408 cases of cryptococcal meningitis [8]. Finally, estimations from a Colombian study reported ~753,523 fungal infections in 2017, where infections by *Candida* species were the most frequent with nearly 600,000 cases [9].

It should be noted that few publications describe cases of IFD carried out in hospital institutions or laboratories. In Chile in 2009, 41 cases of IFD were reported in people with onco-hematologic issues, with aspergillosis being the most common infection, followed by candidiasis and, less frequently, mucormycosis and fusariosis [10]. Subsequently, in 2011, there was an analysis of a five-year (2004–2009), during which 51 episodes of IFD were identified, predominately cases of candidiasis and aspergillosis [4]. Finally, in 2015, the results corresponding to a period of three years were published (2011–2014), during which the study identified a total of 18 patients diagnosed with IFD caused by filamentous fungi, predominantly from genera *Rhizopus* and *Sarocladium* [5]. In Mexico, two studies described the frequency of IFD. The first analyzed 472 cases across 21 years and identified candidiasis as predominant, followed by mucormycosis, cryptococcosis, aspergillosis, and histoplasmosis [11]. The second study described 132 cases identified over a period of 10 years, also predominantly candidiasis, followed by histoplasmosis and cryptococcosis. Other IFD were identified less frequently including coccidioidomycosis and aspergillosis [12]. In Colombia, a cross-sectional, retrospective study in patients with systemic lupus reported 7.5% incidence of invasive fungal infections [13].

Given the lack of information of the epidemiology of IFD in Colombia and the limited laboratory diagnostic capacity at the national level, the objective of this work was to calculate the absolute frequency of IFD adjusted for clinical suspicion, and additionally, to describe the positivity and opportunity of each of the diagnostic assays used in a medical mycology laboratory in Medellín, Colombia.

## 2. Materials and Methods

*Type of study*: A retrospective, descriptive study was conducted with files of patients between18 years and older referred to the Medical and Experimental Mycology Unit at the Corporación para Investigaciones Biológicas (CIB), in Medellín, Colombia, with clinical suspicion of IFD during a 82-month period between March 2009 and December 2015. Information was collected and stored using the specialized clinical laboratory documentation program Vicsoft^®®^.

*Calculation of number of samples*: The total number of samples was calculated by adding the total records of each of the 14 laboratory assays analyzed in this work including: direct examination (wet and nigrosin); special stains (Wright and Grocott-Gomori’s methenamine silver [GMS]); culture and hemoculture for deep mycoses, and API 20C AUX for yeast identification; *Histoplasma* antigen in urine; *Cryptococcus* antigen in serum and cerebrospinal fluid (CSF); *Aspergillus* antigen in serum and bronchoalveolar lavage (BAL); *Aspergillus*, *Paracoccidioides* and *Histoplasma* antibodies in serum and CSF, using agar gel immunodiffusion assay (AGID), complement fixation (CF), or both assays simultaneously. Likewise, *Histoplasma* and *Pneumocystis* specific Polymerase Chain Reaction (PCR), and a universal PCR for fungi (Panfungal PCR) were analyzed [14,15,16].

*Calculation of number of patients*: Because a patient could have multiple samples analyzed by different laboratory assays, this was estimated based on the laboratory assay and total referrals. To generate this number, and in order to eliminate duplicate data, two filters were used: patient identification number and name (total and partial matches were reviewed for the latter).

*Calculation of frequency adjusted for clinical suspicion*: Since there was no variable that specifically differentiated clinical suspicion in the records of orders sent to the laboratory, an indirect estimate of the number of patients suspected for each mycosis was made based on the assays ordered at the time of sample submission. To do so, laboratory assays were classified into two groups: a) assays used for the diagnosis of multiple mycoses (culture, hemoculture and serology for fungi, and panfungal PCR), and b) specific assays (antigen detection for *Histoplasma*, *Aspergillus*, and *Cryptococcus*, and *Histoplasma* and *Pneumocystis* specific PCR). A summary of general and specific assays was performed for each infection, which would later be used as the denominator for calculating the frequency adjusted for clinical suspicion. The combination of general and specific assays for calculating the denominator of each of the IFD is described below:Histoplasmosis: culture and hemoculture for deep mycoses, *Histoplasma* antigen, *Histoplasma* antibodies, *Histoplasma*-specific PCR, and Panfungal PCR.Aspergillosis: culture for deep mycoses, *Aspergillus* antigen, *Aspergillus* antibodies, and Panfungal PCR.Cryptococcosis: culture and hemoculture for deep mycoses, *Cryptococcus* antigen, and Panfungal PCR.Paracoccidioidomycosis: culture for deep mycoses, *Paracoccidioides* antibodies, and Panfungal PCR.Pneumocystosis: methenamine silver stain and *Pneumocystis-*specific PCR.Invasive Candidiasis: culture and hemoculture for deep mycoses, and Panfungal PCR. This infection was diagnosed only in those samples obtained from sites normally considered to be sterile or from biopsy specimens. Broncho-alveolar specimens were excluded.

All samples analyzed by cultures were accompanied by direct examination, and in some cases, depending on the type of sample and clinical suspicion, they were accompanied by negative staining such as nigrosin or special stains (Wright and methenamine silver).

*Definition of cases*: IFD diagnosis was established taking into account the recommendations of the European Organization for Research and Treatment of Cancer/Invasive Fungal Infections Cooperative Group and the National Institute of Allergy and Infectious Diseases Mycoses Study Group (EORTC/MSG) [17]. The total of cases is comprised of the sum of proven and probable cases, defined as follows:Proven case: microorganism isolation by culture or observation with special stains (silver methenamine, Wright, nigrosin, or direct examination), which would indicate the presence of one of the causative agents of IFD, such as: *Pneumocystis jirovecii*, *Histoplasma capsulatum*, *Cryptococcus neoformans*, and *Paracoccidioides brasiliensis*. For the diagnosis of aspergillosis, invasive candidiasis, and other IFD, biological samples from which the isolates were extracted needed to be sterile and may or may not have been accompanied by a direct positive assay. In the case of patients with PCM, two consecutives reactive serologies validated the diagnosis. In the case of patients with histoplasmosis, diagnosis was established by the presence of precipitated bands in the AGID assay, or any titer of anti-*Histoplasma* antibodies in the CF from samples of CSF. Moreover, for patients with cryptococcosis, the presence of a positive antigen assay in CSF or serum is considered as a proven diagnosis of the disease.Probable case: presence of *Histoplasma*- or *Aspergillus*-specific antigens, and/or presence of specific antibodies against *Histoplasma*, *Aspergillus*, and *Paracoccidioides*. To supplement this aim, molecular assays were added as an IFD diagnostic tool. In the case of patients with an *Aspergillus* isolate recovered from non-sterile sites, a probable diagnosis was made based on the growth of the causative agent in at least two culture media, inoculated using aseptic technique at the specimen harvest site, and accompanied or not by a positive direct examination.

*Statistical analysis*: The information analyzed was summarized by calculating absolute and relative frequencies for qualitative variables (gender and laboratory assays used in the diagnosis). For quantitative variables (age and time of result reporting), normality assays and summary measures were performed. For the analysis of differences of the medians of quantitative variables, the Mann-Whitney U test was used. These analyses were performed at a 95% confidence level. Graphs and statistical analyses were conducted using the statistical package GraphPad Prism 5.0^®®^ and Microsoft Excel 2010^®®^.

*Ethical considerations*: The present study was not subject to ethical review, as data were collected through laboratory database review and patients were not interacted with, nor was a manipulation of clinical samples.

## 3. Results

During the period analyzed (March 2009 to December 2015), 33,516 samples were sent to the CIB’s Medical and Experimental Mycology Unit diagnostic laboratory, these samples came from 13,071 patients between 18 years and older with clinical suspicion of IFD. The gender distribution of the study subjects corresponded to 8021 men (61%) and 5050 women (39%), with a median age of 46 years (interquartile range [IR]: 31 to 57 years). Diagnosis of IFD was established in 1425 patients (11%), distributed as follows: 641 with histoplasmosis, 331 with aspergillosis, 239 with cryptococcosis, 111 with pneumocystosis, 60 with PCM, and 48 with invasive candidiasis. The median age of these patients was 42 years (IR: 31–55 years), 1027 of which (72%) were men and 398 (28%) women (Table 1). Analysis of the patients by age according to each IFD showed two groups with a statistically significant difference (*p* < 0.05) between the median ages calculated: one group with a median age of less than 50 years, which included patients with histoplasmosis, cryptococcosis, and pneumocystosis and another group with a median age over 50 years that included patients with PCM, aspergillosis, and invasive candidiasis. When comparing the distribution of age according to gender and diagnosis of IFD, no differences were found amongst patients with pneumocystosis, aspergillosis, or invasive candidiasis. Male patients with histoplasmosis and cryptococcosis were younger than female patients (*p* < 0.001 and *p* = 0.028, respectively). In the case of PCM, the median age in men was 52 years; because PCM alone was diagnosed in just two women, these were insufficient for statistical testing.

The following results were obtained regarding the frequency adjusted for clinical suspicion:

*Histoplasmosis*: 641 of 11,756 patients with clinical suspicion (6%) were diagnosed, 218 of which were proven (34%) and 423 considered to be probable cases (66%) (Table 1). Among the proven cases, 212 had positive cultures (97%); nine of these presented simultaneously on Wright stain, indicating the presence of intracellular yeast compatible with *H. capsulatum*; and the remaining six cases (3%) had reactive serology from CSF (AGID and CF). Probable cases were identified using immunodiagnostic and molecular assays. The 641 cases of histoplasmosis had a total of 835 positive laboratory assays, with varying percentages of positivity, namely: antigenuria for *Histoplasma* n = 318 (38%); antibody detection, n = 255 (31%); fungal cultures, n = 212 (25%); and molecular assays, n = 50 (6%) (Figure 1).

*Aspergillosis*: 331 patients were diagnosed among 10,985 with clinical suspicion (3%), with 19 (6%) classified as proven cases and the remaining 312 patients (94%) as probable cases (Table 1). Probable cases were diagnosed with immunodiagnostic or molecular methods or by isolation of the causative agent in culture from samples taken from non-sterile sites. It should be noted that in this last group, 85 patients presented positive culture from non-sterile samples. The microorganism was also isolated from at least two culture media with pure colonies present at the inoculation site, which were accompanied or not by a positive direct examination. The 331 patients with aspergillosis presented 358 positive assays, of which galactomannan antigen detection was the most frequently positive laboratory assay (n = 184; 51%), followed by cultures for fungi (n = 104; 29%), antibodies detection assays (n = 61; 17%), and finally, molecular assays (n = 9; 3%) (Figure 1). Among the 104 patients in whom the causal agent was isolated in culture, the most frequently isolated species was *Aspergillus fumigatus* (n = 69; 66%), followed by *A. flavus* (n = 25; 24%), *A. terreus* (n = 4; 4%), and less frequently, the following species: *A. niger, A. nidulans, A. versicolor,* and *A. restrictus* (n = 1; 1% each). In two *Aspergillus* isolates (2%), it was not possible to reach species identification. 

*Cryptococcosis*: Diagnosis was established in 239 of 8172 patients with clinical suspicion (3%), all cases were proven (Table 1). The 239 patients with cryptococcosis had 352 positive assays, of which the most frequent was culture (n = 186; 53%), followed very closely by the detection of the capsular antigen of *C. neoformans* (n = 164; 46%), with 93 samples from serum and 71 from CSF. Panfungal PCR was positive in only two patients (1%) (Figure 1). All isolates obtained from the patients were identified as *C. neoformans*.

*Pneumocystis jirovecii pneumonia (pneumocystosis [PCP]):* Diagnosis was established in 111 of 1651 patients with clinical suspicion (7%), with 104 cases proven (94%) and the remaining seven cases probable (6%) (Table 1). In total, 111 of PCP were identified from 138 positive laboratory results, of which methenamine silver stain was the most frequently positive assay (n = 104; 75%), followed by *Pneumocystis jirovecii* specific PCR (n = 34; 25%) (Figure 1).

*Paracoccidioidomycosis (PCM):* Diagnosed in 60 of 10,178 patients with clinical suspicion (0.6%), with 24 cases proven (40%) and 36 probable cases (60%) (Table 1). In the 60 patients with PCM, there were 72 positive assays with the detection of antibodies being the most frequently positive diagnostic method (n = 56; 78%), followed by cultures (n = 14; 19%) and Panfungal PCR (n = 2; 3%) (Figure 1).

*Invasive candidiasis:* Diagnosed in 48 of 7525 patients characterized by clinical suspicion (0.6%), with all cases proven by culture (Table 1). Among the 48 patients there were 52 positive assays, with fungal culture being the most frequently positive diagnostic method (n = 48;92%), followed by Panfungal PCR (n = 4; 8%) (Figure 1). In these 48 cases, the *Candida* species identified through phenotypic carbohydrate assimilation assays, such as API 20C AUX, were: *C. albicans* in 30 cases (63%), followed by *C. parapsilosis* in 10 cases (21%), *C. guilliermondii* in two cases (4%), and with lesser frequency, one case for each of the following species: *C. glabrata, C. krusei, C. norvegensis C. ciferrii, C. kefir,* and *C. magnoliae* (2% each).

Result turnaround time and their respective standard deviations are summarized in Figure 2, where it is noted that antigen detection assays for *Cryptococcus*, *Histoplasma,* and *Aspergillus* yielded results in less time (<24 h to 4 days) in contrast to fungal cultures, which was the assay that required the most time to generate results, according to isolated microorganism, with a turnaround time for results that ranged between 17 and 48 days (Figure 2).

The frequency of cases diagnosed for each IFD during the analyzed period is summarized in Figure 3. During the study period, a 53% increase in cases of infection diagnosed was observed comprehensively, from 150 cases in 2009 to 230 in 2015. Breaking this increase down individually by IFD, histoplasmosis showed an increase in the number of cases per year, going from 52 cases in 2009 to 70 in 2015, noting that in 2014 there was a peak of 133 cases. Aspergillosis had 28 cases in 2009 and 87 in 2015, with an increase of 211%. Likewise, an increase in the number of cases of pneumocystosis was observed, with 13 cases in 2009 and 21 in 2015, with an increase of 62%. The opposite occurred with cryptococcosis, which showed a decrease in the number of cases, from 42 to 29 cases (−30%). PCM cases fluctuated throughout the study period, with the highest incidence occurring during 2012 and 2015. Cases of invasive candidiasis showed a linear trend in the number of diagnoses, starting in 2009 with six cases and ending with nine cases diagnosed during 2015 (Figure 3).

Additionally, other fungal infections were identified by culture and molecular assays less frequently (n = 52) (Figure 1). Through isolation by culture, 44 of the 52 cases (85%) were identified, among which 27 cases of hyaline molds of phylum Basidiomycetes should be highlighted, with 15 of these identified as *Schizophyllum commune*. Other species of hyaline molds identified included *Fusarium* spp with six cases, *Acremonium* spp with three cases, and one case for each of the following agents: *Rhizomucor, Saksenaea vasiformis, Microascus paisii, Paecilomyces variotii, Purpureocillium lilacinum, Scedosporium prolificans, Scedosporium apiospermum*, and *Scedosporium boydii*. Using Panfungal PCR, diagnosis was achieved for a total of eight additional cases (15%): three cases of fusariosis, two of which were produced by species of the *Fusarium solani* complex and one case by the *Fusarium incarnatum-equiseti* complex, two cases of IFD caused by *Rhizopus oryzae*, and one case by each of the following etiological agents: *Schizophyllum commune, Phytium insidiosum,* and *Corynespora cassicola*.

## 4. Discussion

This study retrospectively analyzed laboratory results of 13,071 adult patients with clinical suspicion of IFD, reported in a laboratory specialized in human mycoses in Medellín, Colombia, across a relatively short period of time (82 months). With 1425 laboratory confirmed cases described, this is the largest known sample analyzed in the literature to date. Additionally, this work evaluated the performance of multiple laboratory methods for the diagnosis of different IFD. In our laboratory, the frequency of these diseases is high, with histoplasmosis being the most frequent IFD, followed by aspergillosis, cryptococcosis, pneumocystosis, PCM, invasive candidiasis, and less frequently, IFD caused by other emerging agents including some basidiomycetes and mucorales. Several publications related to IFD based on different methodological designs (prospective, retrospective, multicentric) and different patient populations (onco-hematological, transplant, intensive care unit) have reported for countries in Europe and the Americas a greater frequency of patients with invasive candidiasis and aspergillosis, which is different from the findings observed in our study [2,5,18,19,20,21].

Patients diagnosed with IFD in our study had a median age of 42 years and were more frequently male. However, when analyzing the median age, a difference was observed between the group of patients with IFD associated with advanced HIV disease (histoplasmosis, cryptococcosis, and pneumocystosis) and the group of patients with aspergillosis, candidiasis, and PCM. In the former, the median age corresponded to younger patients, with 50% of cases between 30 and 50 years. This pattern is similar to the age distribution of people living with the HIV in Colombia, in which the most frequently affected group is concentrated between 20 and 40 years of age [22]. In the case of patients with aspergillosis and candidiasis, it is well known how age increase the risk to development these infections, either by deteriorating the response capacity of the host’s immune system, or by the appearance of other chronic conditions that facilitate the development of these diseases [18,23]. In the case of PCM, in which the median age of those affected was greater than 50 years, it was previously known that this disease is more frequent in patients over 30 years of age, with predominance among males. Also recognized, is the long period of latency from infection to clinically manifested disease (more than 30 years), as well as the protective hormonal factor of 17β-Estradiol in women [24,25].

Cultures were the chief diagnostic method used for achieving proven diagnoses, despite having sensitivities ranging between 11% and 80%. These ranges can depend on the type of sample, the immunological status, the moment of diagnosis of the infection, and the limitations inherent to the time of reporting the results. These are, however, vital for determination of in vitro sensitivity to antifungal agents [5,26].

Histoplasmosis and aspergillosis were the IFD that involved a greater number of laboratory assays to reach their diagnosis. Immunological assays, detection of antigens and antibodies, were the most frequently assays involved. It is important to emphasize that depending on the disease clinical form, these assays can reach sensitivity values greater than 90% [27,28,29,30].

In the diagnosis of histoplasmosis, it is important to note that, although most laboratory assays appear to be efficacious, their sensitivity and specificity vary depending on the clinical presentation of the disease [31]. Antigen detection assays show a higher performance in patients with a diagnosis of progressive disseminated histoplasmosis (PDH), in which the sensitivity of this method can range from 80% to 95% [28,32]. In our laboratory, this technique has been used since 2011, which could explain the significant increase in the frequency of diagnosis of this disease [28,33]. As for antibody detection assays (AGID and CF), these were also very useful for the diagnosis of histoplasmosis cases, and even helped in the diagnosis of cases tested from CSF samples. These assays have a better performance in the diagnosis of acute forms (75% to 95% sensitivity) and chronic forms of the disease (70% to 90% sensitivity) [32]. Fungal cultures can reach up to 85% sensitivity in the diagnosis of PDH, but the delivery of results requires several weeks of analysis (between 2 to 6 weeks) [34]. Finally, molecular assays also contributed in a good extent in the detection of probable cases of histoplasmosis. In our laboratory, a nested PCR was used, which identifies the gene that codes for a 100 KD protein with 95% sensitivity and 92% specificity, which is why it is very useful in a large variety of clinical samples for the detection of *H. capsulatum* DNA, since it is more sensitive than culture, and therefore, a useful tool in endemic regions [14].

Diagnosis of aspergillosis involved in the majority of cases, the detection of *Aspergillus* galactomannan antigen, demonstrating its utility as a diagnostic tool. This is confirmed by some meta-analyses, which have shown that the average sensitivity and specificity of the assay is around 70% and 90%, respectively. Therefore, its greater diagnostic utility has been described in onco-hematological patients with a higher risk of invasive aspergillosis [26,27]. Additionally, the detection of anti-*Aspergillus* antibodies in serum by the AGID technique can contribute in the diagnosis of aspergillosis in the allergic bronchopulmonary form [26,35]. The cultures were also a useful tool for the diagnosis of this disease. However, it is necessary to have experience and protocols for the interpretation of the cultures, since it is possible to isolate *Aspergillus* as a result of contamination of the samples. In this study, panfungal PCR with sequencing was less used for the detection of the cases. It is important to say, that the use of sterile samples is requested to perform this assay. Another advantage of molecular methods is their ability to identify the species of the genus *Aspergillus* associated with IFD [26,36].

The diagnosis of cryptococcosis in most cases, used culture and capsular antigen detection simultaneously. It is important to highlight that antigen detection techniques, such as latex particle agglutination (LA) and lateral flow immunochromatographic assay (LFA), have sensitivity and specificity values greater than 95% in serum, plasma, and CSF [37,38,39,40]. Additionally, the ability to detect circulating antigen in asymptomatic patients has been reported. Currently, the WHO recommends routine screening in serum or plasma of HIV positive patients with less than 100 CD4 T cells/μL, which allows timely initiation of appropriate antifungal treatment and reduce the risk of complications and mortality in these patients [41]. This type of point-of-care (POC) assays are easy to perform, affordable, rapid (results in less than an hour), and have the potential to significantly improve the early diagnosis of cryptococcosis [42].

Pneumocystosis was the diagnosis made in the majority of cases using methenamine silver stain, with molecular assays also proving significant. In addition, it has been documented that PCR is more sensitive than the staining methods [43]. The nested PCR validated in our laboratory has shown a sensitivity of 86% and a specificity of 98% when using deep respiratory samples [15]. This method shows excellent diagnostic value and higher negative predictive value than microscopy, sufficient to confirm or rule out the diagnosis of this disease in high-risk patients [43]. While in general, diagnosis of pneumocystosis were made using diagnostic methods that generated rapid results, molecular assays were of great value, since the low fungal burden in non-HIV patients can lead to false negative results in staining techniques. This sensitivity can also be improved by nested PCR that uses two rounds of amplification, increasing the diagnostic sensitivity [43].

Diagnosis of PCM was made mainly by using antibody detection assays. It is noteworthy that the AGID is a simple, inexpensive method to perform that yields result quickly. Serological methods can have a high diagnostic value, mainly when the AGID and CF are performed simultaneously, reaching sensitivities between 75% and 98% [44]. Additionally, recommendations of the EORCT/MSG define that two reactive serologies in a patient confirm the diagnosis of this disease. To a lesser extent, patients were diagnosed as proven cases through the use of cultures and direct examination. It is important to bear in mind that *Paracoccidioides brasiliensis* is a slow-growing microorganism, thus, the incubation period of the media may require several weeks of analysis.

In our laboratory, all cases of invasive candidiasis were diagnosed as proven cases using cultures, but this method showed a prolonged delay before yielding results. This work encourages revisiting the need to optimize the delivery time of results for this mycosis, implementing improvements such as the use of chromogenic media and periodic review of these cultures, allowing quicker identification of mixed yeast infections of the *Candida* genus. It is also necessary to implement faster and more sensitive methodologies, such as molecular methods based on PCR principles, mass spectrometry, or nanotechnology. There is also a strong need to develop alternative and complementary techniques to the conventional mycological diagnosis that are affordable and timely. [26,36]. Additionally, it is important to note that contradicted what is described in the national and international literature, in our laboratory, the diagnosis of invasive candidiasis was the least frequent among the mycoses analyzed, which; the literature states that invasive candidiasis is the most frequent IFD [2,18]. In our case this can be explained by two reasons, the first being that our laboratory is more specialized and has more tools for the diagnosis of endemic mycoses (histoplasmosis and PCM) and opportunistic infections (aspergillosis, cryptococcosis, and pneumocystosis). The second reason may be due to the fact that many hospital institutions carry out the diagnosis of invasive candidiasis in their own laboratories, mainly institutions that have intensive care units or other specialized services. For this reason, these medical centers only refer to specialized laboratories those cases with a complex diagnosis. Therefore, in this study the actual frequency of this mycosis is likely underestimated.

In our study, histoplasmosis, aspergillosis, and pneumocystosis showed an increasing trend in the number of cases, possibly due to the fact that during the time in which this study was conducted, new methods were implemented for the detection of antigens and molecular diagnostic assays, methods that are characterized for having higher sensitivity values compared with conventional methods [14,15,26,27,28,29,30,33,34,36]. In most studies, reports of patients with invasive candidiasis have been reported to be more frequent than what we found in our study [4,5,10,11,12]. This may be explained by the fact that our laboratory is not affiliated with a hospital, where the majority of cases of invasive candidiasis arise. Rather, our laboratory is a center for the study of endemic mycoses, which has had funding to develop research projects that are applied both to the development and technological transfer of new diagnostic methods and strategies of education [28,45].

In the case of cryptococcosis, the decrease in the number of cases reflects the global epidemiological trend. This decrease may be influenced by more adequate control of HIV infection, which has resulted from the use of more effective antiretroviral therapy and prophylactic treatments with fluconazole, which reduce the risk of developing this disease [46,47].

Paracoccidioidomycosis, on the other hand, is a mycosis with low incidence in Colombia, with the number of cases being relatively few when compared to countries such as Brazil [25,48].

In the case of other mycoses caused by opportunistic fungi such as *Fusarium*, *Schizophyllum*, *Scedosporium*, and *Mucor*, among others, the frequency was not very high in our study, but there has been a notable increase in these diseases in recent years due to the increase in the risk population including immunocompromised patients. These mycoses are associated with high mortality rates, making a rapid diagnosis essential. In our study, with regard to these mycoses, molecular assays had great value, since they allowed reaching a diagnosis at the species level, which allows a specific treatment [49,50].

This work presented various limitations, including the inability to access data related with patient’s histopathology, and clinical and epidemiological information related to risk factors of these patients and outcomes. It should be noted that our study does not report prevalence estimates or incidences of these fungal infections, and therefore, cannot be compared to other studies where these estimates are evaluated. Our work is based on the description of the frequency of the major IFD handled by our laboratory and on the description of the positivity of the different diagnostic methods used. Therefore, the findings regarding the frequency of IFD cannot be applied to the adult Colombian population. It is also important to note that, when comparing this research with other reported studies, many of them based in the diagnosis of IFD using conventional diagnostic methods, which do not provide very high sensitivity values, such as the immunodiagnostic and molecular methods used in this investigation [4,5,10,11,12].

## 5. Conclusions

In conclusion, this work describes the frequency of IFD that was diagnosed in a local laboratory in Colombia and demonstrates how the use of multiple diagnostic methods positively impacts the number of patients diagnosed with these IFD. One should keep in mind that although conventional techniques continue to be of great importance, they must be accompanied by immunological and molecular methods, which offer faster, more sensitive, and specific results. Therefore, implementation of educational strategies will allow medical staff to establish clinical suspicion and give them the ability to perform appropriate diagnostic assays, allowing for more targeted and efficacious treatment that ultimately impacts morbidity and mortality of these fungal diseases.

## Figures and Tables

**Figure 1 jof-06-00039-f001:**
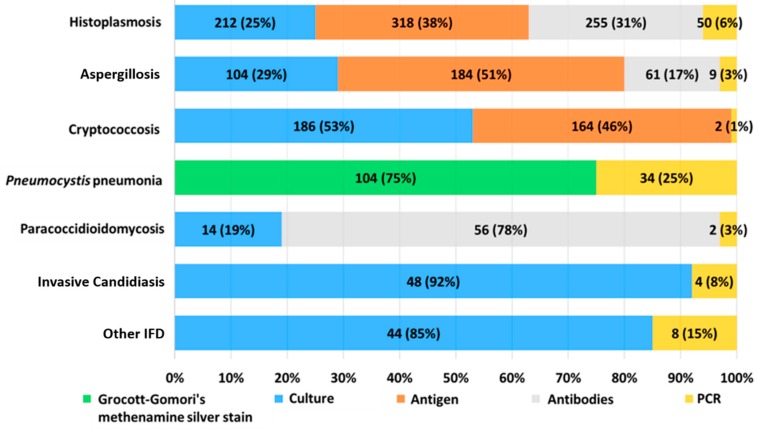
Frequency of positivity among laboratory assays used in the diagnosis of invasive fungal diseases.

**Figure 2 jof-06-00039-f002:**
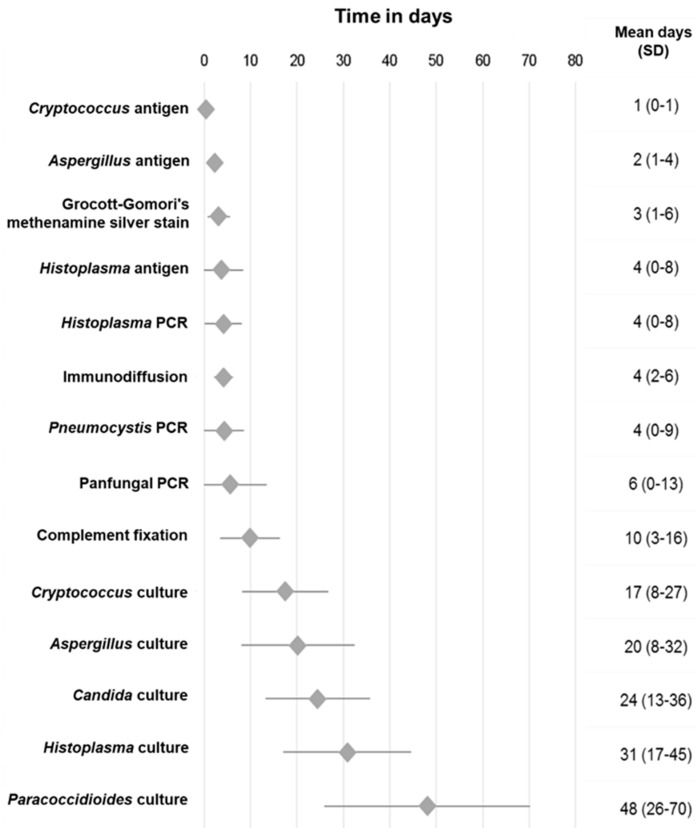
Analysis of result turnaround time, according to laboratory methods used in the diagnosis of Invasive fungal diseases.

**Figure 3 jof-06-00039-f003:**
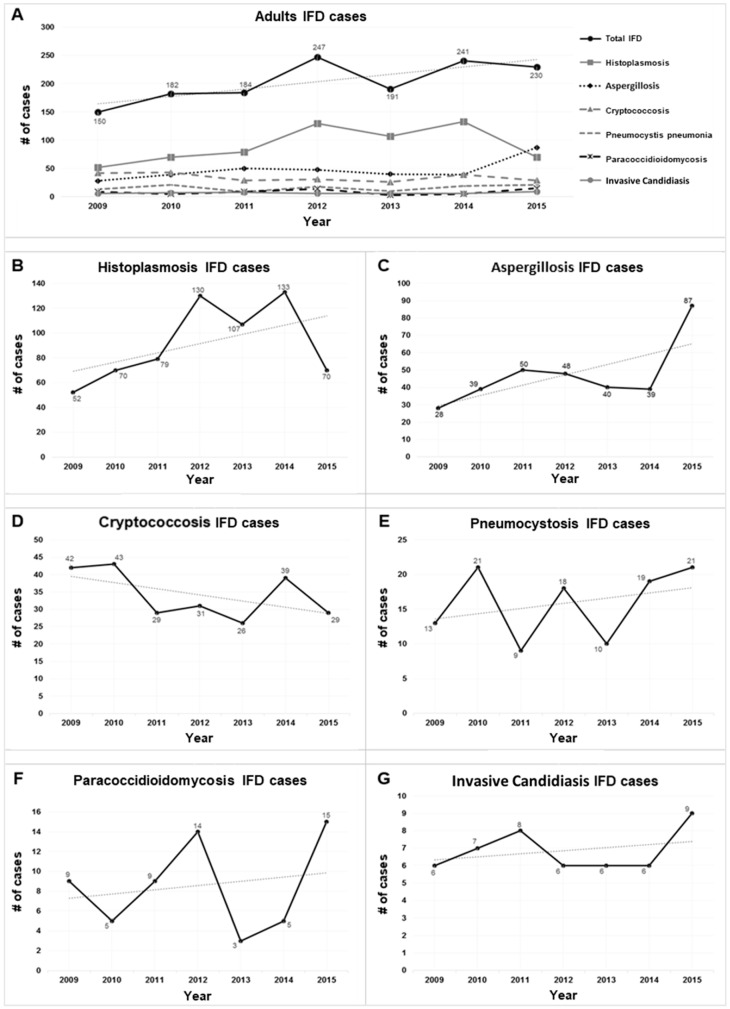
Annual cases of Invasive fungal diseases (2009–2015): (**A**) All Invasive fungal diseases (IFD). Breaking this down individually by IFD: (**B**) histoplasmosis cases; (**C**) aspergillosis cases; (**D**) cryptococcosis cases; (**E**) pneumocystosis cases; (**F**) paracoccidioidomycosis cases; (**G**) invasive candidiasis cases.

**Table 1 jof-06-00039-t001:** Patients diagnosed with invasive fungal disease (IFD).

Diagnosed with IFD	# of Positive Cases/ Denominator (% of Positive Cases)	Proven/Probable Cases	Median Age Years	Male/Female Ratio
**Histoplasmosis**	641/11,756 (6%)	218 (34%)/423 (66%)	38 (IQR:30–49)	3:1
**Aspergillosis**	331/10,985 (3%)	19 (6%)/312 (94%)	54 (IQR:39–65)	1:1
**Cryptococcosis**	239/8172 (3%)	239 (100%)	39 (IQR:30–51)	3:1
**Pneumocystosis**	111/1651 (7%)	104 (94%)/7 (6%)	39 (IQR:33–51)	3:1
**Paracoccidioidomycosis**	60/10,178 (0.6%)	24 (40%)/36 (60%)	52 (IQR:45–59)	29:1
**Invasive Candidiasis**	48/7525 (0.6%)	48 (100%)	53 (IQR:40–63)	3:1
**Total IFD ***	**1425/13,071 (11%)**	**652 (46%)/778 (54%)**	**42 (IQR: 31–55)**	**3:1**

(IQR): Interquartile range of age (values: 25th and 100th percentiles); (#) number (*) four cases with coinfections: Case 1: Histoplasmosis/cryptococcosis/aspergillosis; Case 2: Histoplasmosis/cryptococcosis; Case 3: Cryptococcosis/invasive candidiasis; Case 4: Histoplasmosis/Paracoccidioidomycosis.

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
