# Peer review of "Frequency of Invasive Fungal Disease in Adults: Experience of a Specialized Laboratory in Medellín, Colombia (2009–2015)"

_jof, 2020, doi:10.3390/jof6010039_

Round 1

Reviewer 1 Report

This is a well-written study and would be of interest to readers. This was surprising that why Candida agents were the last cause of invasive fungal diseases, while it is generally well-accepted that Candida agents the most prevalent cause of IFD, almost everywhere (with exceptions of course). This matter is discussed in the discussion but not reflected in the abstract. Therefore, authors are encouraged to highlight this controversy and underestimation in the abstract.

The introduction is well-written, but it would be better if the authors provide some information regarding methods used for the detection/identification of genera of fungi identified in this study. This will lead to shortening the discussion, too. This should not include more than 4 lines.

Lines 269-282, this line is better to be presented in the introduction so that readers can readily obtain some information about the epidemiology of IFD in South American countries. But, please shorten this, using the commonalities found in the epidemiology of IFD in these countries (maybe 3-4 lines) (for instance authors can state that Candida agents were the most cause of IFD in x,y, and z, etc.,). Afterward, authors can use this information to substantiate the importance of conducting this study, which fills the gap in the knowledge of IFD in Colombia.

All Candida isolates were identified by API, therefore better to indicate this in the methods section dealing with identification of invasive candidiasis agents.

Although there is a wealth of information presented, there is no indication of mortality in this study. It would be important to indicate the all-cause mortality for each genus found if data is available or mention this as a limitation if not available.

The burden of fungal diseases is previously reported in JOF and led by Dr. Denning, (https://www.ncbi.nlm.nih.gov/pmc/articles/PMC6023354/) and another group explored the IFD in patients suffering from systemic lupus erythematosus (https://www.ncbi.nlm.nih.gov/pubmed/29536803). Please use these studies as your references and compare your data with this and mention that what is the added value of the current study with the one led by Dr. Denning.

Author Response

Reviewer 1

This is a well-written study and would be of interest to readers. This was surprising that why Candida agents were the last cause of invasive fungal diseases, while it is generally well-accepted that Candida agents the most prevalent cause of IFD, almost everywhere (with exceptions of course). This matter is discussed in the discussion but not reflected in the abstract. Therefore, authors are encouraged to highlight this controversy and underestimation in the abstract.

R/ Thanks you for your comment. We modified manuscript abstract, we added the following statement “Frequency of IFD in this study was atypical compared with other studies, probably this is a result of the single laboratory-site analysis

The introduction is well-written, but it would be better if the authors provide some information regarding methods used for the detection/identification of genera of fungi identified in this study. This will lead to shortening the discussion, too. This should not include more than 4 lines.

R/ We added the following text in the manuscript introduction “Diagnosis of IFD is done using many different laboratory assays. For most of these diseases the gold standard for diagnosis is based on conventional laboratory assays using culture or histopathology (including special stains), these assays present several limitations, been the principals the need for high-level laboratory infrastructure for cultures handling (biosecurity level 3) and highly trained laboratory staff. These assays also presented variable analytical performance, and long turn-around time for laboratory assay results (6). Other alternatives for IFD include assays for the detection of specific antibodies, antigens, and fungal DNA, these alternative assays present higher analytical performance and lower turn-around time for laboratory assay results compared with conventional laboratory assays (6).

Lines 269-282, this line is better to be presented in the introduction so that readers can readily obtain some information about the epidemiology of IFD in South American countries. But, please shorten this, using the commonalities found in the epidemiology of IFD in these countries (maybe 3-4 lines) (for instance authors can state that Candida agents were the most cause of IFD in x,y, and z, etc.,). Afterward, authors can use this information to substantiate the importance of conducting this study, which fills the gap in the knowledge of IFD in Colombia.

R/ We moved the text from the discussion to the introduction as you suggested.

All Candida isolates were identified by API, therefore better to indicate this in the methods section dealing with identification of invasive candidiasis agents.

R/ We indicated the use of the API 20C AUX in the material and methods section as you suggested.

Although there is a wealth of information presented, there is no indication of mortality in this study. It would be important to indicate the all-cause mortality for each genus found if data is available or mention this as a limitation if not available.

R/ Unfortunately we have not access to data about patients’ outcomes. We added this in the study limitations section.

The burden of fungal diseases is previously reported in JOF and led by Dr. Denning, (https://www.ncbi.nlm.nih.gov/pmc/articles/PMC6023354/) and another group explored the IFD in patients suffering from systemic lupus erythematosus (https://www.ncbi.nlm.nih.gov/pubmed/29536803). Please use these studies as your references and compare your data with this and mention that what is the added value of the current study with the one led by Dr. Denning.

R/ Thanks, we added these two references in the manuscript introduction. Compared data of these two studies with our study is difficult, this principally due the significant differences in studies design and population.

Reviewer 2 Report

Based in a retrospective database, this study shows the frequency of invasive endemic mycoses due to dimorphic fungi and also the frequency of invasive opportunistic fungal infections caused by molds and yeasts in a single center in Medellin, Colombia. Only adult patients were evaluated.

Please review the meaning of “neglected diseases," lines 51-53. The World Health Organization (WHO) acknowledges the term neglected diseases as a symptom of poverty and disadvantage. To date, only mycetoma and chromoblastomycosis are officially recognized as “neglected diseases” although paracoccidioidomycosis and sporotrichosis are in the process of recognition. In the other hand, IFI like invasive aspergillosis and invasive candidiasis are not truly neglected but only sub diagnosed.

One of the study’s limitation is that the database included only the results of biologic samples like sera, CNF, sputum, BAL, secretions and aspirates but not histopathologic tissue sections. This may explain the lower frequency of paracoccidioidomycosis for example. Could you please add a comment on this important issue?

Author Response

Reviewer 2

Based in a retrospective database, this study shows the frequency of invasive endemic mycoses due to dimorphic fungi and also the frequency of invasive opportunistic fungal infections caused by molds and yeasts in a single center in Medellin, Colombia. Only adult patients were evaluated.

Please review the meaning of “neglected diseases," lines 51-53. The World Health Organization (WHO) acknowledges the term neglected diseases as a symptom of poverty and disadvantage. To date, only mycetoma and chromoblastomycosis are officially recognized as “neglected diseases” although paracoccidioidomycosis and sporotrichosis are in the process of recognition. In the other hand, IFI like invasive aspergillosis and invasive candidiasis are not truly neglected but only sub diagnosed.

R/ Thanks for your comment, we modified the text as follows: “Globally, mycoses are not considered as priory public health problem and are non-notifiable, which contributes to their under registration”

One of the study’s limitation is that the database included only the results of biologic samples like sera, CNF, sputum, BAL, secretions and aspirates but not histopathologic tissue sections. This may explain the lower frequency of paracoccidioidomycosis for example. Could you please add a comment on this important issue?

R/ We added this limitation in the manuscript discussion.

Round 2

Reviewer 1 Report

Authors have addressed the previous concerns fully and the paper seemed to be ready for publication.